# Locoregional and Surgical Treatment of Single-Nodule Hepatocellular Carcinoma Recurrence After Liver Transplantation: A Systematic Review and a Meta-Analysis

**DOI:** 10.3390/cancers17091501

**Published:** 2025-04-29

**Authors:** Marco Maria Pascale, Camilla Marandola, Francesco Frongillo, Erida Nure, Salvatore Agnes

**Affiliations:** 1General Surgery and Organ Transplant Unit, Fondazione Policlinico Universitario “A. Gemelli” IRCCS, 00168 Rome, Italy; 2Faculty of Medicine, Università Cattolica del Sacro Cuore, 20123 Milan, Italy

**Keywords:** hepatocellular carcinoma, recurrence, liver transplantation, surgical treatment, locoregional treatment

## Abstract

Liver transplantation (LT) is a curative treatment for patients with hepatocellular carcinoma (HCC), especially those with advanced liver disease. However, HCC can return after LT, with recurrence rates between 15–20% within the first two years. Managing small, localized recurrences effectively is key to improving patient outcomes. This study reviewed previous research comparing two treatments for HCC recurrence after LT: surgical resection and locoregional therapies (LRT). After following strict selection guidelines, ten studies from 2009 to 2024 were analyzed. Results showed that patients who had surgery had better outcomes than those treated with LRT. Specifically, one-year overall survival (OS) was higher in the surgery group (71% vs. 62%), and one-year disease-free survival (DFS) was also better (60% vs. 54%). It’s important to note that patients receiving LRT tended to have more aggressive cancer features before their transplant, such as microvascular invasion and higher alpha-fetoprotein levels. In conclusion, while surgery offers better survival chances, treatment decisions must consider each patient’s tumor features and liver condition. Future guidelines that include new treatments like immunotherapy and targeted therapies will help improve care for patients facing HCC recurrence after liver transplantation.

## 1. Introduction

Liver transplantation (LT) is the most effective treatment option for patients with hepatocellular carcinoma (HCC), especially in cases of advanced liver disease or neoplasms unsuitable for other local treatments like resection or ablation [1]. As one of the most prevalent malignancies globally, HCC typically arises from chronic liver conditions, such as hepatitis B and C, alcoholic liver disease, or metabolic-associated fatty liver disease (MAFLD), and is complicated by high morbidity and mortality [2]. Given these challenges, LT offers an effective dual solution: it not only removes the liver tumor but also replaces the damaged organ, reducing the risk of cancer recurrence associated with a damaged hepatic environment [3].

Nevertheless, transplantation offers a curative solution for patients with advanced disease or multifocal tumors within set criteria, which are aimed at selecting patients with a lower risk of recurrence.

Since the Milan criteria were introduced in the 1990s, patient selection for LT in HCC cases has become more precise. These criteria, which prioritize single tumors no larger than 5 cm or up to three tumors no larger than 3 cm each, aim to select candidates with a favorable prognosis [4]. Subsequent criteria, like the UCSF criteria, have sought to expand the pool of transplant candidates while maintaining acceptable post-transplant outcomes, offering an effective therapeutic pathway for patients who may not have qualified under Milan standards [5].

Despite the benefits, post-LT HCC recurrence remains a challenge, occurring in approximately 15–20% of cases within the first two years after surgery [6]. Standard recurrence prevention involves rigorous selection through the adoption of the aforementioned criteria, neoadjuvant treatments, and postoperative immunosuppressive management.

Understanding and managing recurrence is crucial to improve the outcomes [7]. The mechanisms driving HCC recurrence post-LT are multifactorial. Recurrence may result from residual microscopic disease not detected or removed at the time of LT or from metastatic dissemination that had occurred prior to the procedure. In some cases, the persistence of underlying oncogenic risk factors, such as viral hepatitis or metabolic syndrome, may contribute to the formation of new tumors [8].

The management of HCC recurrence after LT presents significant challenges. Due to the immunosuppressive environment required to prevent graft rejection, typical immune responses against cancer are weakened, complicating therapeutic options. Despite these challenges, several strategies have shown promise in managing both disseminated and localized recurrences [9].

The treatment strategies for a single-nodule HCC recurrence post-liver transplant can generally be categorized into surgical and local therapies. Surgical resection, which involves the removal of the recurrent tumor through surgery, is often considered when the patient is in good health and the lesion is deemed resectable. This approach can potentially achieve a clear margin of excision, which is associated with a lower risk of further recurrence. Surgical resection, however, is an invasive procedure that carries certain risks, especially in post-transplant patients who are often immunosuppressed and may have compromised liver function due to the cumulative effects of medications and underlying comorbidities. The invasiveness of surgery, coupled with the potential risk to the transplanted organ, often limits the applicability of resection in this population [10].

Locoregional therapies offer alternatives to resection, particularly in cases where the tumor is small, solitary, or where surgical resection would involve excessive risk. Transarterial Chemoembolization (TACE) delivers chemotherapy directly to the tumor site, reducing systemic exposure. Radiofrequency Ablation (RFA) and Microwave Ablation (MWA) use high-frequency radio waves to destroy cancer cells.

These therapies are less invasive than surgery and can often be repeated if recurrence occurs. However, they may not provide as definitive a solution as surgical resection, especially in cases where complete tumor removal is achievable through surgery [11].

The choice between surgical and local therapies in the setting of a single HCC recurrence after an LT requires a nuanced approach that accounts for multiple factors.

Our work aims to perform a literature review on the oncological and short-term survival outcomes in patients with localized recurrence of HCC after LT who are undergoing surgery or LRT.

## 2. Methods

This systematic review was conducted following the Preferred Reporting Items for Systematic Reviews and Meta-Analyses (PRISMA) Statement criteria. The prospective review protocol was registered on PROSPERO with the following ID: 635506. If a reviewer was present in the authorship of a paper in evaluation, the decision to include that study was undertaken by other members of the review group.

As this was a review, ethics approval was not required.

### 2.1. Study Eligibility Criteria

Studies were included according to the following eligibility criteria: (1) Prospective or retrospective studies of HCC recurrence after LT; (2) publication within the last 15 years (2009–2024); (3) a liver malignancy histologically characterized as HCC, (4) an LT as prior surgical treatment, (5) a diagnosis of unifocal HCC recurrence, (6) a surgical treatment of the HCC recurrence as intervention; (7) a comparator of locoregional treatment (LRT), based on radiofrequency ablation (RFA), transarterial chemoembolization (TACE) or transarterial radioembolization (TARE), (8) one or more patient outcomes (1 year overall survival, 1 year disease-free survival).

We excluded papers without survival outcomes reported specifically. If studies presented overlapping patient populations, only the most up-to-date cohorts were included.

Randomized controlled trials (RCTs) and nonrandomized studies (nonrandomized controlled trials, interrupted time series, controlled before-and-after studies, cohort studies) were eligible for inclusion. Animal and preclinical studies were excluded, as were opinion manuscripts, conference abstracts, trial protocols, and grey literature.

### 2.2. Search Strategy and Selection of Studies

A systematic electronic search was performed on PubMed and Google Scholar up to November 2024.

The electronic search strategy was conducted by using a combination of the following MeSH terms and free text words: “Liver Transplantation* OR Carcinoma Hepatocellular*/pathology OR Carcinoma, Hepatocellular*/surgery OR Carcinoma, Hepatocellular/drug therapy* OR Liver Neoplasms*/surgery OR Liver Neoplasms/pathology OR Carcinoma, Hepatocellular/drug therapy* OR Liver Transplantation*/adverse effects AND Neoplasm Recurrence, Local OR Neoplasm Recurrence, Local/mortality* OR Graft Survival*”.

Two authors (M.M.P. and C.M.) independently extracted data from each included study.

### 2.3. Endpoints

The outcomes assessed in our work focused on fundamental indicators of transplant and oncological efficacy. The data limitations across cohorts, the high early mortality in this population, and the relevance of short-term survival in guiding clinical decision-making for recurrent HCC prompted us to choose a short but informative timing in daily clinical practice. These outcomes included the 1-year overall survival (OS) and 1-year disease-free survival (DFS). The data for each outcome were extracted directly, as reported in the source studies, without reclassifications of the rates.

### 2.4. Selection Process

Title and abstract screening and full-text screening were conducted by two researchers (M.M.P. and C.M.) in a standardized, independent manner using the Rayyan software program (https://rayyan.ai, accessed on 26 April 2025) [12]. If there was any disagreement, it was disclosed at the end of the independent individual evaluation and discussed until reaching a final decision. A manual search was also conducted on papers autonomously selected by the researchers because of their pertinence to the query.

In addition, reference lists of the included articles were manually searched in order to retrieve any possible full-length papers that could be included.

### 2.5. Data Collection

General information on the included papers (i.e., study design, year of publication, country, duration of the study) and data related to patients (i.e., age and gender, etiology underlying the HCC, inclusion in Milan criteria at time of transplant, histological and pathological characterization of the recurrences) were extracted from single papers and collected into a customized table.

### 2.6. Study Quality Assessment

The study quality assessment was performed throughout the modified Newcastle-Ottawa scale by 2 authors (M.M.P. and C.M.) [13] (Table 1). The methodological quality of the included studies was dishomogeneous. Seven articles out of seventeen reached a score equal to or higher than 7, such as Bodzin et al. [14], which, with a score of 8, represented the article with the highest methodological quality above those selected. Others, on the contrary, had an elevated risk of bias [15,16]. Nevertheless, we included their evidence in our qualitative analysis because we considered it as not neglectable. Furthermore, we tried to minimize the selection risk of bias by choosing strictly the inclusion criteria, such as including only studies performed on a population of patients whose histologically diagnosed HCC had been already treated by LT and had recurred. Additionally, we only included studies that described the treatment implemented for the recurrence and its efficacy in terms of survival. On top of that, we struck out case reports, conference abstracts, trial protocols, and grey literature, as well as any form of literature in languages other than English. Due to the limited availability of literature on the topic, the articles we selected may have a dishomogeneous description of the outcomes, a short follow-up, or patients could have received a combination of more than one treatment. The shortcomings mostly concerned the comparability and the outcomes domains: in fact, studies had different follow-up periods, some of them were not precise in the characterization of the recurrence features and the response to treatment, while others displayed a paucity of details when describing the population of the study and their clinical and pathological background.

### 2.7. Statistical Analysis

Pooled treatment effects for single-proportion outcomes were analyzed using a random-effects generalized linear mixed model with a 95% confidence interval (CI). Heterogeneity was evaluated with the tau-squared and I^2^ test by Higgins and Thompson. Heterogeneity was considered significant in the case of an I_2_ result of at least 40%. Subgroup analysis by type of treatment was performed in the case of 1 or more studies reporting an outcome. Subgroup analysis included pooling subgroup outcomes using a random-effects model, and subgroup effects were compared using the Q test, with a significant result defined as a *p*-value < 0.05. Jamovi^®^ version 2.3.28 was used for statistical analysis.

## 3. Results

### 3.1. Study Selection

The electronic search provided 1174 records (PubMed: 968 papers, Scholar: 206 papers). Among them, 62 duplicates were detected and struck out. Then, 1034 papers were selected for full-paper evaluation. The manual search retrieved three additional articles, which underwent a full-text evaluation, providing a total of 1037 reviewed papers. Twelve articles fulfilled the inclusion criteria [14,15,16,17,18,19,20,21,22,23,24,25]. The subsequent study quality assessment rejected two articles. Ten articles, thus, were included in qualitative and quantitative synthesis [14,15,17,18,19,20,21,23,24,25]. The selection process is reported as a flow diagram, following the PRISMA guidelines, in Figure 1.

### 3.2. Study Characteristics and Summary of Results

This systematic review includes ten retrospective cohort studies and a prospective study.

General information on the included papers (i.e., study design, year of publication, country, number of patients, study period) is reported in Table 2.

The included study spanned from 2000 to 2024. Most of the studies (*n* = 6) came from China. One study was from Germany, one from France, two from the USA, and one from Korea.

The multifocal intrahepatic recurrence (more than three nodules) was present in the liver graft in the majority of the samples described by the studies we analyzed (in five of the studies, more than half of the population had more than three intrahepatic nodules at the time of recurrence). In eight of the studies, the majority of patients displayed both intra- and extrahepatic localizations of the recurrence.

For the sites of extrahepatic recurrence, in eight of the studies, the most frequently affected organ was the lung. Other common sites of recurrence were bone (in six studies), kidney (in one study), adrenal gland (in six studies), and brain (in two studies).

In all of the studies, the main outcomes were overall survival and disease-free survival after HCC recurrence was used as the principal outcome. Furthermore, in four of the studies, there was also a retrospective analysis of the characteristics generally displayed by patients with HCC recurrence suitable for surgical treatment. Six of the studies compared the efficacy of each of the three main therapeutic strategies (surgery, chemotherapy, and LRT), one only focused on the efficacy of microwave ablation, one on chemotherapy, and one on radiofrequency ablation.

A total of 5665 patients affected by HCC underwent LT. Among them, 791 patients displayed recurrence of the disease, either intra- or extrahepatic.

Overall, among these patients, 189 patients benefited from surgical treatment of the recurrence, whereas 273 were treated with nonsurgical approaches, be they systemic or locoregional. In the details, the LRT was found in 148 patients. Eleven patients underwent chemoembolization, 26 patients received a TACE treatment, 24 patients benefitted from RFA, and 87 from an MWA approach.

### 3.3. Overall Survival

Data for approximately 1-year OS were available in 10 studies for 265 patients (Figure 2).

A total of 189 patients underwent surgical treatment, while 76 patients underwent LRT.

The overall survival at 1 year in the surgical group was higher, as opposed to the LRT group (71% vs. 62%, *p* = 0.038).

### 3.4. Disease-Free Survival

Data for approximately 1-year DFS were available in nine studies for 258 patients (Figure 3).

A total of 182 patients underwent surgical treatment, while 76 patients underwent LRT.

In the surgical group, the DFS rate was 60%, compared to a DFS rate of 54% in the LRT group (*p* = 0.042).

### 3.5. Predictive Clinical-Pathological Variables Predictive of Outcomes

We compared the pre-LT and LT clinical-pathological characteristics of the patients with post-LT HCC recurrence who underwent surgical treatment or LRT.

Patients outside Milan criteria prior to LT were 42% in the LRT group and 18% in the surgical group (*p* = 0.031).

About the histological features of the HCC on the whole liver at the time of LT, the rate of the poor differentiation aspect was 22% in the surgical group and 61% in the LRT group (*p* = 0.018).

The microvascular invasion was more present in the HCC nodules of patients of the LRT group (38%) than in patients of the surgical group (21%).

Furthermore, the pre-LT analysis of oncological markers revealed a distinct pattern between the two groups. Specifically, the median alpha-fetoprotein level was more significant in the LRT group (342 ng/mL, IQR 128–560) compared to the surgical group (180 ng/mL, IQR 95–261) (*p*-value = 0.042).

## 4. Discussion

HCC is a major indication for LT in patients with cirrhosis and advanced liver disease. Despite stringent selection criteria for LT in cases of HCC, the recurrence of HCC after LT is a significant concern, with reported rates ranging from 10% to over 30%, depending on pre-LT tumor characteristics and treatment modalities [26]. A significant proportion of patients experience intrahepatic or extrahepatic HCC relapse post-transplant. Managing HCC recurrence is challenging due to the immunosuppressed state of the patient, which promotes tumor progression. Intrahepatic recurrences are particularly challenging due to the compromised liver function that often accompanies them [27]. Consequently, managing these recurrences requires a thorough understanding of treatment options available to optimize patient outcomes. Among the treatment options, surgical resection and LRT stand out as the most effective strategies for localized recurrences.

We performed a review of ten studies and 265 patients to compare outcomes of surgical treatment and LRT in patients with HCC recurrence after LT. The main results showed that the 1-year disease-free survival and 1-year overall survival rates were better in the surgically treated cohort. However, patients treated with LRT had more advanced pre-LT HCC characteristics in terms of clinical-radiological features (with Milan criteria), histopathological differentiation and microvascular invasion, and alpha-fetoprotein concentration.

Surgical resection represents a potentially curative approach for managing recurrent HCC after LT. Studies indicate that surgical resection can lead to improved survival rates among selected patients [28,29]. The ability to perform a resection depends heavily on the extent of recurrence, the location of the tumor, and the underlying function of the liver graft post-transplantation. Factors such as the number of lesions and the presence of extrahepatic metastases significantly influence eligibility for surgical management [30].

Recent studies have reported favorable outcomes for patients who underwent salvage hepatectomy after the failure of locoregional therapy [28]. Minagawa et al. demonstrated that repeat hepatectomy for locally recurrent HCC resulted in satisfactory oncological outcomes comparable to those seen with initial surgeries [28]. Patients undergoing salvage hepatectomy exhibited lower recurrence rates and better long-term survival than those limited to palliative care or untreatable by locoregional procedures.

However, surgical treatment is associated with complications and risks inherent to surgery, particularly in a hepatic context [30]. The liver graft, already compromised by the initial disease and transplantation process, may not tolerate further surgical interventions as effectively as a native liver does. Thus, careful patient selection and multidisciplinary evaluation are essential for optimizing outcomes.

LRT, which includes ablation techniques, transarterial chemoembolization (TACE), and radiotherapy, serves to target HCC with localized intervention while preserving liver function [31]. These treatments can often be repeated and used in combination with surgical options to manage recurrences effectively.

RFA is a widely used locoregional treatment for HCC, particularly in patients with small tumors. The technique is minimally invasive and can be repeated as necessary, which is particularly advantageous for managing recurrent disease [31]. Studies have demonstrated that RFA can effectively ablate small tumors and yield good local control rates, particularly when combined with other therapies [32]. Outcomes following RFA for recurrent HCC suggest that this approach may be less invasive than reoperation while still providing comparable survival benefits in selected patients [33].

TACE is another prominent locoregional therapy that combines the delivery of chemotherapy directly to the tumor with embolization to restrict blood flow [34]. During recurrence management, TACE may be employed for larger or multifocal lesions that are unsuitable for surgical excision, potentially increasing eligibility for subsequent transplantation by downstaging tumors.

LRT has also been studied for its potential role in downstaging HCC to meet the Milan criteria, allowing patients to become eligible for transplantation despite initially presenting beyond these strict thresholds [30,35]. Evidence suggests that TACE combined with locoregional therapies enhances survival rates compared to TACE alone, particularly in recurrent settings [36].

The choice between surgical and locoregional treatments for HCC recurrence following LT is complex, influenced by tumor burden, liver function, and prior treatments. Comparisons of OS and RFS between these modalities suggest that surgical options tend to offer better survival rates in select patient cohorts [26,30]. Foerster et al. noted similar survival rates for patients undergoing transplants for initially unresectable HCC and those undergoing salvage transplantation or resection following recurrence, underscoring the importance of timing and patient selection in maximizing surgical outcomes [29].

Contrastingly, locoregional treatments provide significant advantages through repeatability and lower overall risk profiles compared to extensive resection surgeries or transplantation. Patients who may not tolerate general anesthesia or have poor liver function may benefit more from locoregional approaches, which can also result in significant symptom relief and tumor control without the need for more extensive surgical interventions [31,37].

One of the major challenges in treating recurrent HCC after LT is the balance between eradicating the tumor and maintaining hepatic function. Surgical interventions inherently carry risks of further liver parenchyma loss, which can lead to postoperative liver failure in patients with limited functional reserve [30]. The presence of underlying cirrhosis and other chronic liver diseases complicates the prognosis significantly.

In contrast, locoregional therapies are generally more conservative, preserving liver function while allowing for continuous monitoring of disease progression. However, systemic therapies and locoregional treatments have yielded mixed results, particularly in the context of extrahepatic metastases, where surgical options may present limited benefits due to multifocal disease and a higher propensity for recurrence [38].

Deciding between surgical and locoregional treatment requires a case-by-case analysis, factoring in tumor characteristics, liver function, and the patient’s overall clinical situation [39]. A multidisciplinary approach involving hepatologists, surgical oncologists, and transplant surgeons is critical to tailor the best treatment pathway for each individual.

While LRT has seen advancements, including the potential for combination regimens that may enhance outcomes, the relative predictability of surgical interventions frequently underscores its perceived superiority in specific situations [36].

A fundamental limitation of our study is the absence of data about the concomitant or subsequent administration of chemotherapeutic drugs. Sorafenib and Regorafenib showed solid results also in HCC recurrence in patients who already undergone LT [40,41]. Immunotherapy and targeted agents are the other notable absence in our work. Immune checkpoint inhibitors like nivolumab and ipilimumab have shown promising results [42]. Anyway, all the systemic therapies exhibit more data in advanced HCC settings but are under-evaluated in post-transplant scenarios, despite some evidence suggesting tolerable side effects and the potential for meaningful outcomes in this cohort [43]. The absence of robust data on the use of systemic therapies poses a significant gap in the management strategies available for a single recurrence of HCC after LT.

Future research should aim to explore the integration of systemic therapies into the treatment regimen for HCC recurrence after liver transplantation. Combinations of locoregional therapies with immunotherapy may enhance management outcomes by addressing both local and systemic tumor activity [44,45]. Furthermore, establishing clear protocols for monitoring and early detection of recurrence using advanced imaging modalities, along with a deeper exploration of tumor biology and patient-specific factors, will be crucial in refining treatment approaches [46,47]. Addressing these gaps could lead to more personalized treatment strategies and potentially improved survival rates for transplant recipients facing HCC recurrence.

## 5. Conclusions

The management of recurrent HCC after LT presents significant challenges, necessitating an individualized approach that considers patient characteristics, tumor biology, and existing liver function. Both surgery and LRT play crucial roles in the therapeutic landscape, offering unique benefits and limitations. Understanding the nuances of these treatment options will ultimately lead to better decision-making and improved outcomes for patients facing HCC recurrences following LT.

## Figures and Tables

**Figure 1 cancers-17-01501-f001:**
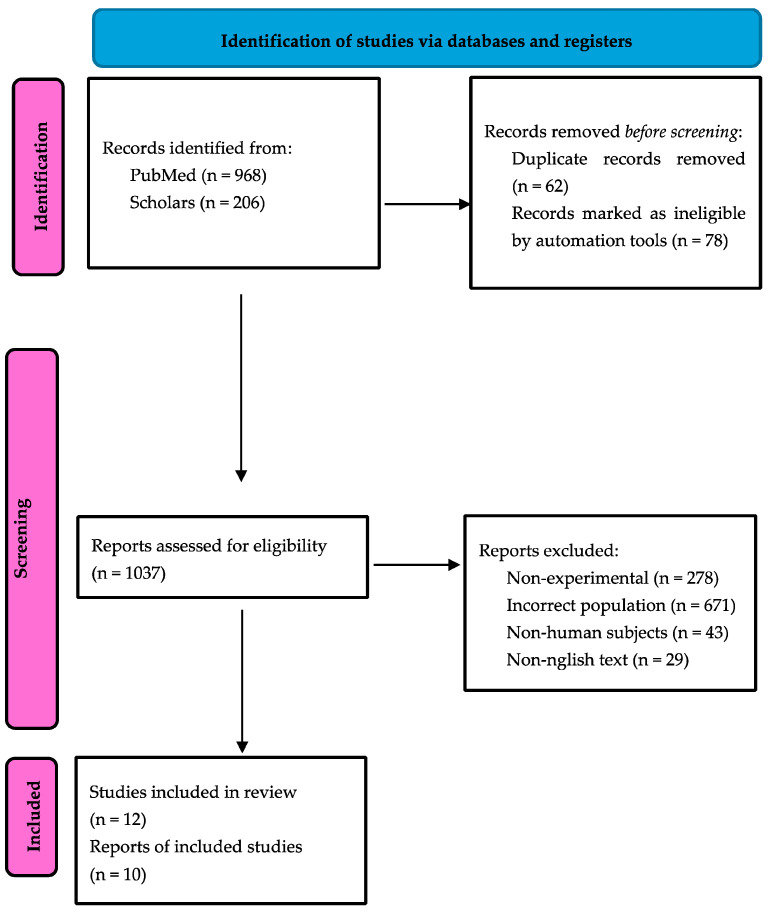
PRISMA flow diagram.

**Figure 2 cancers-17-01501-f002:**
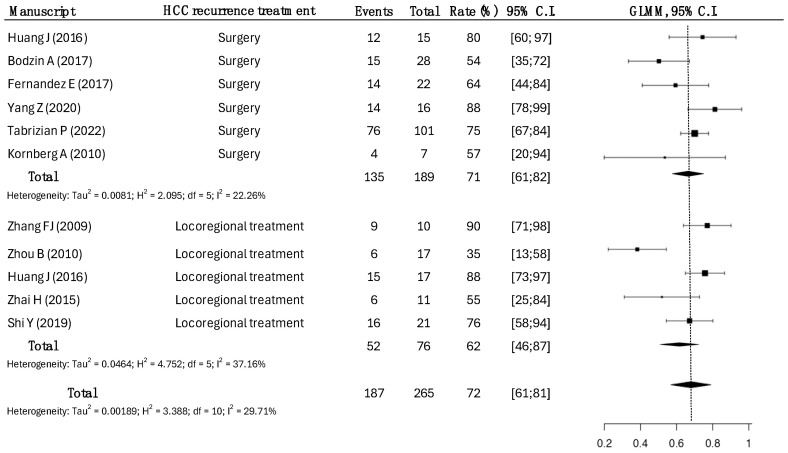
One-year overall survival rates across types of HCC recurrence treatments. Surgical treatments had a survival rate of 71% (95% CI: 61–82), compared to the 62% (95% CI: 46–87) in LRT, with statistical significance (*p* = 0.0038) [14,15,17,18,19,20,21,22,23,25].

**Figure 3 cancers-17-01501-f003:**
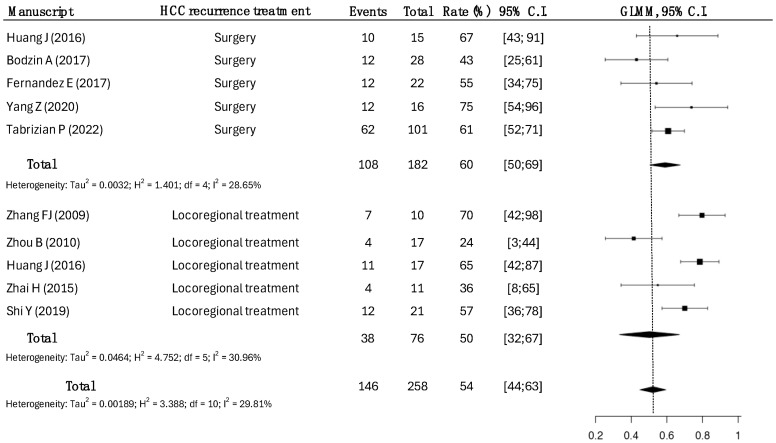
One-year disease-free survival rates according to the treatment modalities of HCC recurrence. Surgical strategy had a disease-free survival rate of 60% (95% CI: 50–69), while the one-year disease-free survival rate in LRT was 50% (95% CI: 32–67), with a slight statistical significance (*p* = 0.048) [14,15,17,18,19,20,21,23,25].

**Table 1 cancers-17-01501-t001:** Newcastle-Ottawa scale of the papers selected. On the left side are the categories of the scale. Every star stands for a point on the scale. In gray, the papers excluded after quality assessment.

Studies	Representativeness of the Exposed Cohort	Selection of the Non Exposed Cohort	Ascertainment of Exposure	Demonstration That the Outcome of Interest Was Not Present at the Start	Comparability of the Cohorts on the Basis of Design or Analysis	Assessment of the Outcome	Was Follow Up Long Enough for Outcomes to Occur	Adequacy of Follow Up Cohorts	Overall Score
	**Selection**	**Comparability**	**Outcome**	
Tabrizian P (2022) [17]									7
Zhou B (2010) [18]									6
Zhang FJ (2009) [15]									6
Shi Y (2019) [19]									6
Zhai H (2015) [20]									7
Fernandez-Sevilla E (2017) [21]									7
Kim HR (2011) [16]	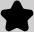	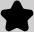	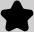	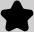		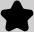			5
Bodzin AS (2017) [14]									8
Kornberg A (2010) [22]									7
Yang Z (2020) [23]									7
Ekpanyapong (2020) [24]	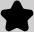	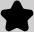	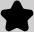	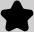		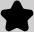			5
Huang J (2016) [25]									7

**Table 2 cancers-17-01501-t002:** Main characteristics of the papers analyzed in the review.

References	First Author	Year of Publication	Country	Study Period	Number of Transplants	Mean age of Recipients	Recipient Male Sex (n,%)	alfa-FP at LT (ng/mL)	Pathology at LT—Number of Nodules (Mean)	Pathology at LT—Size of Largest Nodule (cm) (Mean)	Pathology at LT—Poor Differentiation (%)	Pathology at LT—Microvascular Invasion	Follow Up Protocol	Time to Recurrence (Months) (Mean)	HCC Recurrence	Mean Follow Up Period (Months)
[15]	Zhang FJ	2009	China	2004–2008		46.9	9, 90	5 pts above 200	-	-	-	-	alfa-FP + CT/3 mo	10.9	10	4 to 44
[18]	Zhou B	2010	China	2003–2007	241	47.9	27, 96	-	1.95	3.85	-	-	alfa-FP + CT/US/3 mo	-	71	14.5
[22]	Kornberg A	2010	Germany	1994–2007	60	56.5	46, 76	-	4	6	8.3	20	alfa-FP + US/3 mo; CT/6 mo	23	16	5 to 180
[20]	Zhai H	2015	China	2008–2014		52.5	11, 100	-	-	3.1	-	-	-	-	11	5 to 33
[25]	Huang J	2015	China	1997–2012	486	42 below 50 yo	57, 73	60 pts above 400	-	-	67.9	16.6	-	25.3	78	39.7
[21]	Fernandez-Sevilla E	2017	France	1991–2013	493	55	61, 87.1	15 pts above 100	3	4	-	57.1	alfa-FP + US/3 mo	17	70	78
[14]	Bodzin E	2017	USA	1984–2014	857	57	-, 77	mean 47.2	3	3.8	33	59.4	CT + alfa-FP/6 mo	15.8	106	60
[19]	Shi Y	2019	China	2010–2017	52	49.5	47, 90.4	-	-	2.4	-	-	CT or MRI/6 mo	22.1	52	31.4
[23]	Yang Z	2020	China	2016–2018	293	48	59, 90.6	-	3	-	53.1	23.4	alfa-FP + US/CT/3 mo	-	64	10.2
[17]	Tabrizian P	2022	US	2001–2015	2645	59.9	2028, 76.7	mean 20.5	1	2	-	-	-	-	330	55.3

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
