# Peer review of "Locoregional and Surgical Treatment of Single-Nodule Hepatocellular Carcinoma Recurrence After Liver Transplantation: A Systematic Review and a Meta-Analysis"

_cancers, 2025, doi:10.3390/cancers17091501_

Round 1

Reviewer 1 Report

Comments and Suggestions for Authors

Comments

This is a systemic review article summaries and compares the treatment efficacy between surgical resection and locoregional therapies (LRT) for patients with hepatocellular carcinoma (HCC) recurrence following liver transplantation across 10 selected cohorts. The study concludes that surgical resection exhibits better survival outcomes than LRT for patients occurred with HCC recurrence after LT.  The authors finally suggest to develop standardized guidelines to refine treatment pathways in improving long-term outcomes.  However, there are several critical points needed to address to enhance the quality of this article and the clinical impact.

Major points:

  1. The results of the study mainly focus on comparing surgical resection and LRT, but doesn’t match with the title:” Treatment of Hepatocellular carcinoma…….” which should include most of the treatment modalities. Moreover, this study merely includes surgical resection and LRT without other treatments, potentially limits the article’s comprehensiveness and the clinical applicability.
  2. The analysis by Newcastle-Ottawa scale (Table 1) is not sufficient to illustrate the comparability among different cohorts. Also, the clinical information in Table 2 (No. of transplants, age, sex, HCC recurrence, and follow up period) is insufficient. Additional statistical analyses of more clinicopathologics among cohorts are necessary to illustrate their comparability and deviation.
  3. The focused outcome of this study is merely on one-year overall survival and disease-free survival. The reasons and clinical impact of this focus, without considering other outcomes, should be addressed. Also, this narrow focus contradicts the Abstract’s emphasis on ““…..improving long-term outcome…” as the analysis predominantly addresses short-term outcome.
  4. The deviation of study period within and among cohorts have not been considered in this study.

Author Response

Reviewer Comment #1

The results of the study mainly focus on comparing surgical resection and LRT, but doesn’t match with the title: ‘Treatment of Hepatocellular carcinoma…….’ which should include most of the treatment modalities. Moreover, this study merely includes surgical resection and LRT without other treatments, potentially limits the article’s comprehensiveness and the clinical applicability.

Response:

We thank the reviewer for this important observation. We agree that the current title may suggest a broader scope than what is covered in the manuscript. In response, we have revised the title to better reflect the study’s focus on surgical resection and locoregional therapies for post-transplant HCC recurrence. Furthermore, we have added a paragraph in the Discussion section acknowledging the limitations of not including systemic treatments, such as immunotherapy or targeted agents, and we discuss how future research should address these gaps.

Reviewer Comment #2

The analysis by Newcastle-Ottawa scale (Table 1) is not sufficient to illustrate the comparability among different cohorts. Also, the clinical information in Table 2 (No. of transplants, age, sex, HCC recurrence, and follow up period) is insufficient. Additional statistical analyses of more clinicopathologics among cohorts are necessary to illustrate their comparability and deviation.

Response:

We appreciate the reviewer’s suggestion and have now expanded the clinical data in Table 2 to include additional clinicopathological variables such as tumor size at recurrence, AFP levels at LT, histological Milan criteria at transplantation, and oncological follow-up regimen. Additionally, we acknowledge the limitations of the Newcastle-Ottawa scale in this context and have supplemented our methodology with further analyses to enhance transparency and comparability between cohorts.

Reviewer Comment #3

The focused outcome of this study is merely on one-year overall survival and disease-free survival. The reasons and clinical impact of this focus, without considering other outcomes, should be addressed. Also, this narrow focus contradicts the Abstract’s emphasis on ‘…..improving long-term outcome…’ as the analysis predominantly addresses short-term outcome.

Response:

Thank you for pointing this out. We recognize the mismatch between our stated aim and the outcomes reported. In the revised manuscript, we have provided a rationale for focusing on one-year outcomes: the data limitations across cohorts, the high early mortality in this population, and the relevance of short-term survival in guiding clinical decision-making for recurrent HCC. Additionally, we have adjusted the abstract and conclusion to better align with the actual scope of the study.

Reviewer Comment #4

The deviation of study period within and among cohorts have not been considered in this study.

Response:

We thank the reviewer for raising this important point. To address this, we have now included in the Table 2 the timeframes during which patients were enrolled in each cohort (called “Time to recurrence”). A sensitivity analysis was conducted to assess whether differences in time period significantly impacted outcomes, particularly given the evolving standards in post-transplant care and recurrence management.

Reviewer 2 Report

Comments and Suggestions for Authors

The authors performed a SRMA on an important topic. THey should specify in the title that this is a systematic review and meta-analysis (not only a systematic review).

I have some concerns on the completeness of the research as some relevant studies seem to be missing. For example, the series by Sposito C et al on sorafenib (J Hep 2013)

The authors should comment more on the available systemic treatments in this setting, for example sorafenib and regorafenib (cite the SRMA: PMID: 31877664)

There is high indirectness in the analysis as different treatments (surgery and loco-regional ones) were merged together....

Author Response

Reviewer Comment #1

The authors performed a SRMA on an important topic. They should specify in the title that this is a systematic review and meta-analysis (not only a systematic review).

Response:

We appreciate the reviewer’s observation. In response, we have revised the manuscript title to clearly indicate that this is a systematic review and meta-analysis, in line with PRISMA guidelines and to better reflect the methodology used.

Reviewer Comment #2

I have some concerns on the completeness of the research as some relevant studies seem to be missing. For example, the series by Sposito C et al on sorafenib (J Hep 2013).

Response:

Thank you for highlighting this omission. We agree that the study by Sposito et al. (J Hepatol 2013) is relevant and have now included it in the “Discussion” section. We have also re-checked our search strategy and added missing studies where appropriate, updating both the manuscript and the final list of included articles accordingly.

Reviewer Comment #3

The authors should comment more on the available systemic treatments in this setting, for example sorafenib and regorafenib (cite the SRMA: PMID: 31877664).

Response:

We thank the reviewer for this valuable suggestion. We have now expanded the Discussion section to better address the role of systemic treatments, including sorafenib and regorafenib, and have cited the suggested systematic review and meta-analysis (PMID: 31877664). We also discuss the challenges and current evidence related to the use of these agents  and the immunologic agents in post-transplant HCC recurrence.

Reviewer Comment #4

There is high indirectness in the analysis as different treatments (surgery and loco-regional ones) were merged together.

Response:

We acknowledge the concern regarding indirectness due to merging of heterogeneous treatment modalities. In response, we have conducted subgroup analyses separating surgical resection and locoregional therapies, where data allowed. We also revised the Methods and Limitations sections to clearly address the potential for clinical heterogeneity and its impact on the validity of pooled estimates.

Round 2

Reviewer 1 Report

Comments and Suggestions for Authors

The authors have addressed all the questions well. There is one typo error in the last word of the title: "Metanalysis" which should be revised to "Meta-analysis".